# Reduced Nitrogen Rate with Increased Planting Density Facilitated Grain Yield and Nitrogen Use Efficiency in Modern Conventional *Japonica* Rice

Tianyao Meng [1], Xi Chen [1], Jialin Ge [2], Xubin Zhang [2], Guisheng Zhou [1], Qigen Dai [2] and Huanhe Wei [2,*]

[1] Joint International Research Laboratory of Agriculture and Agri-Product Safety, Institutes of Agricultural Science and Technology Development, The Ministry of Education of China, Yangzhou University, Yangzhou 225009, China; 007126@yzu.edu.cn (T.M.); 15148856902@163.com (X.C.); gszhou@yzu.edu.cn (G.Z.)
[2] Jiangsu Co-Innovation Center for Modern Production Technology of Grain Crops, Jiangsu Key Laboratory of Crop Cultivation and Physiology, Research Institute of Rice Industrial Engineering Technology, Yangzhou University, Yangzhou 225009, China; gejialin@foxmail.com (J.G.); zxb19961206@163.com (X.Z.); qgdai2000@126.com (Q.D.)
* Correspondence: 006931@yzu.edu.cn

**Abstract:** The past three decades have seen a pronounced development of conventional *japonica* rice from the 1990s, although little information is available on changes regarding grain yield and nutrient use efficiency during this process. Nine conventional *japonica* rice released during the 1990s, 2000s, and 2010s were grown under a reduced nitrogen rate, with increased planting density (RNID) and local cultivation practice (LCP) in 2017 and 2018. The rice from the 2010s had 3.6–5.5% and 7.0–10.1% higher ($p < 0.05$) grain yield than the 2000s and the 1990s, respectively, under RNID and LCP. The harvest index contributed more to genetic yield gain from the 1990s to the 2000s; whereas from the 2000s to 2010s, yield increase contributed through shoot biomass. Genetic improvement increased total nitrogen (N), phosphorus (P), and potassium (K) accumulation, and their use efficiencies. The rice from the 2010s showed a similar grain yield, whereas the 1990s and 2000s' rice exhibited a lower ($p < 0.05$) grain yield under RNID relative to LCP. RNID increased N, P, and K use efficiencies, particularly the N use efficiency for the grain yield (NUEg) of the 2010s' rice, compared with LCP. For three varietal types, RNID increased the panicles per m$^2$, the filled-grain percentage, and the grain weight ($p < 0.05$) while decreasing spikelets per panicle of the 2010s' rice. Compared with LCP, RNID reduced non-structural carbohydrate (NSC) content and shoot biomass, at heading and maturity, while increasing the remobilization of NSC and the harvest index, especially for the 2010s' rice. Our results suggested the impressive progressive increase in grain yield and nutrient use efficiency of conventional *japonica* rice since the 1990s in east China. RNID could facilitate grain yield and NUEg for modern conventional *japonica* rice.

**Keywords:** conventional *japonica* rice; genetic improvement; grain yield; nitrogen use efficiency; reduced nitrogen rate; increased planting density





## 1. Introduction

Rice is a staple crop worldwide, so increasing its productivity is a principal strategy for ensuring food security [1]. Genetic improvement is recognized as a key driver in enhancing rice yields [2]. A modeling study estimated that variety improvement contributed 74% to the total increased rice production after the 1980s in China [3].

Genetic yield gain has been extensively studied across rice-growing countries [4–6]. Most existing literature suggests a pronounced yield increase over years; for example, Anzoua et al. [6] estimated a 2.55 g m$^{-2}$ year$^{-1}$ yield gain of rice varieties released from 1905 to 1988 in Northern Japan. Improvements for high yield were associated with an enlarged sink size potential [6,7], an increased biomass accumulation and/or harvest index [8], an optimized plant morphology [4,9], and a coordinated source-sink balance [10,11].

The proportion of cultivation area between conventional *japonica* rice and hybrid *indica* rice has undergone a dramatic transformation during the last three decades in Jiangsu, east China [12]. The occupation of conventional *japonica* rice and hybrid *indica* rice planting area has shifted from 10% and 90% in the 1990s, to 90% and 10% in the present day. Such a pronounced varietal renewal has contributed greatly to the boosting of rice production in this region, which has helped increase the provincial average yield (t ha$^{-1}$) from 7.5 in the 1990s, to 8.6 today, with the highest yield level per unit area among major rice-production provinces in China [13]. Apart from grain yield, there is increasing interest concerning crop nutrient use efficiency during genetic improvement [14]. The impacts of genetic improvement on nutrient use efficiency were studied in wheat [15,16], maize [17,18], barley [19,20], and cotton [21]. However, the responses of nutrient use efficiency to genetic improvement varied with the specific crops and experimental conditions. For instance, nitrogen (N), phosphorus (P), and potassium (K) use efficiencies were increased by 20%, 23%, and 24% for cotton varieties released over 33 years in Australia [21]; whereas there was no clear trend of N use efficiency among barley released from 1902 to 1988 in Finland [20]. At present, few studies have been reported concerning the changes in the grain yield and nutrient use efficiency of conventional *japonica* rice released after the 1990s in Jiangsu, east China.

The rice production system is typically characterized by a high grain yield and a relatively lower use efficiency of the nutrient, especially for N elements in Jiangsu. Many endeavors, such as reduced N rates with an increased planting density (RNID), were proposed for the optimization of grain yield and N use efficiency (NUE) in rice production. Compared with local cultivation practices (LCPs), RNID could maintain grain yield and NUE by raising panicles per m$^2$ [22,23], radiation interception and canopy photosynthesis potential [24,25], as well as leaf area index, shoot biomass accumulation [26], and root growth [27]. Moreover, RNID is also considered environmental friendly for mitigating greenhouse gas emissions [28,29]. Such above studies on RNID for the facilitation of grain yield and NUE were mostly conducted on hybrid *indica* rice in south and central China; whether RNID could facilitate simultaneous grain yield and NUE of conventional *japonica* rice in Jiangsu still awaits investigation. It is also interesting to explore the responses of grain yield and nutrient use efficiency to conventional *japonica* rice bred in different periods after the 1990s, under RNID.

Here, nine conventional *japonica* rice released during the 1990s, 2000s, and 2010s were grown under LCP and RNID in the fields. The present study had two main objectives: (1) to evaluate the changes and differences in grain yield and use efficiency of nutrients for conventional *japonica* rice released since the 1990s; (2) to investigate the possibility of RNID facilitating changes in grain yield and nutrient use efficiency of conventional *japonica* rice in Jiangsu.

## 2. Materials and Methods

### 2.1. Experimental Site, Rice Variety, Field Design, and Crop Establishment

In 2017 and 2018, field experiments were carried out at Yangzhou University, Jiangsu, east China. Generally, the rice experienced similar temperatures (24.5 °C vs. 24.6 °C), sunshine hours (1105 h vs. 1152 h), and rainfall (816.3 mm vs. 859.1 mm) during the growing period from May to October, across two years. There was an amount of 0–20 cm soil in the experimental field, which had a sandy loam texture, and contained 18.3 g organic carbon kg$^{-1}$, 1.5 g total N kg$^{-1}$, 31.8 mg Olsen P kg$^{-1}$, and 85.1 mg available K kg$^{-1}$.

The field experiment was a 2 × 9 (two cultivation treatments and nine rice varieties) factorial design with 18 treatment combinations. Each of the treatments had three replications in a completely randomized block design and gave a total of 54 plots. Each experimental plot covered 26 m$^2$ (6.5 m × 4 m). Cultivation treatments were LCP (300 kg ha$^{-1}$ N; 30 hills per m$^2$, 25 cm × 13.3 cm), and RNID (255 kg ha$^{-1}$ N; 36 hills per m$^2$, 25 cm × 11 cm). The N management and planting density in the LCP was designed following local

technicians, based on previous studies [30,31]. The specific information on the nitrogen management and planting density of LCP and RNID can be seen in Table 1.

**Table 1.** The specific information on nitrogen management and planting density of two cultivation treatments.

| Cultivation Treatment | Nitrogen Application Rate and Timing (kg ha$^{-1}$) | | | | | Planting Density | | |
|---|---|---|---|---|---|---|---|---|
| | Total Rate | One Day before Transplanting | One Week after Transplanting | Panicle Initiation | Penultimate Leaf Appearance | Hill Spacing | Seedling per Hill | Seeding Number per m$^2$ |
| LCP | 300 | 90 | 90 | 60 | 60 | 25 cm × 13.3 cm | 4 | 120 |
| RNID | 255 | 76.5 | 76.5 | 51 | 51 | 25 cm × 11.0 cm | 4 | 144 |

LCP, local cultivation practice; RNID, reduced nitrogen rate with increased planting density.

Nine conventional *japonica* rice varieties were grown in this field experiment and were classified into three types, the 1990s, 2000s, and 2010s, according to their year of release. The rice varieties from the 1990s were Wuyujing 3, Zaofeng 9, and Wuyunjing 8. The rice varieties from the 2000s were Wujing 15, Nanjing 44, and Yangjing 4038. The rice varieties from the 2010s were Nanjing 9108, Zhendao 18, and Wuyunjing 31. The selection of these rice varieties was based on cultivation by local institutes and popularity among farmers. For example, Wuyujing 3 variety was the most popular variety in the 1990s in East China and its cultivation area was up to 6.1 million hectares by end of 2019 (www.ricedata.cn, accessed on: 12 June 2021). In recent years, Nanjing 9108 and Wuyunjing 13 have become very popular and widely planted rice varieties, owing to their superior palatability [13]. These rice varieties shared a similar growth pattern; the total growth period ranged from 152 d to 157 d in this study. The detailed information on the year of release, cross information, and cumulative planting area of these rice varieties is listed in Table 2.

**Table 2.** Detailed information on year of release, cross information, and cumulative planting area of rice varieties.

| Varietal Type | Variety | Year of Release | Cross Information | Cumulative Planting Area (×10$^4$ hm$^2$) |
|---|---|---|---|---|
| 1990s | Wuyujing 3 | 1992 | Zhongdan 1/79-51 × Zhongdan 1/Yangjing 1 | 608 |
| | Zaofeng 9 | 1997 | Wufujing/Zhongdan 1 × Nonglin 205 | 104 |
| | Wuyunjing 8 | 1999 | Xiangnuo 9121/Jia 48 × Bing 815 | 88 |
| 2000s | Wujing 15 | 2004 | Zaofeng 9/Chunjiang03jing × Wuyunjing 7 | 88 |
| | Nanjing 44 | 2007 | Nanjing 38 | 45 |
| | Yangjing 4038 | 2008 | Zhenxiang 24/Wuyunjing 8 × Chang 9363 | 22 |
| 2010s | Nanjing 9108 | 2013 | Wuxiangjing 14 × Guandong 194 | 111 |
| | Zhendao 18 | 2013 | Zhendao 99 × Wuyunjing 7 | 17 |
| | Wuyunjing 31 | 2015 | Yun 2608 × Fan 103 | 10 |

Source: http://www.ricedata.cn, accessed on: 12 June 2021.

Pre-germinated rice seeds were sown in seedbeds on 23 May and transplanted with four seedlings per hill on 12 June, during the rice cultivation season of 2017 and 2018. N was applied as urea (46% N) at the ratio of 3:3:2:2, 1 d before transplanting, 1 week after transplanting, panicle initiation, and the penultimate leaf appearance, respectively. P (180 kg ha$^{-1}$ as $P_2O_5$) and K (135 kg ha$^{-1}$ as $K_2O$) fertilizers were applied as a basal dressing. The irrigation regime adopted in the field experiment followed alternate wetting and drying cycles. Chemical controls of pest, disease, and weed were performed following local recommendations.

### 2.2. Sampling and Measurement

At jointing, heading, and maturity, plants of four hills were collected to determine the leaf area index (LAI), shoot biomass, non-structural carbohydrate (NSC) content, and total nutrient (N, P, and K) accumulation. The sampled plants were subdivided into stems

and leaves at jointing, and stems, leaves, and panicles at heading and maturity. LAI was measured through a Leaf Area Meter (LI-3100C, Lincoln, NE, USA). The collected parts were placed well in Kraft paper bags, and the shoot biomass was recorded after 80 h of oven-drying at 75 °C. After weighing, the dried samplings were ground using a Wiley mill (Thomas Scientific, Swedesboro, NJ, USA) for determination of the NSC content and nutrient (N, P, and K) analysis. The NSC content and nutrient concentrations in the rice plants were assayed, following the method of Nakano et al. [32] and Inthapanya et al. [33].

The rice varieties shared a similar growth pattern; thus, the soil–plant analysis development (SPAD) values of nine rice varieties were determined at 12, 24, and 36 days after heading (DAH), as well as for leaf photosynthetic rate at 15, 30, and 45 DAH. The SPAD-502 plus was used for the measurement of the SPAD values of the top three leaves from 14:30 h to 16:00 h. The photosynthetic rate of the flag leaf was performed through three photosynthetic instruments (LICOR-6400, Lincoln, NE, USA), and measurements were performed from 9:30 h to 11:00 h under sunny conditions.

Plants of two hundred representative hills in each plot, excluding border plants, were harvested for the determination of grain yield (expressed at 14% moisture) at maturity. In each plot, plants of one hundred representative hills were collected for measuring panicles per m$^2$, spikelets per panicle, filled-grain percentage, and grain weight. The filled-grain percentage was calculated as the ratio of filled grains, selected in a salt solution with a gravity of 1.06 [34].

*2.3. Formula Calculation and Statistical Analysis*

NSC remobilization reserve = ((NSC content in the stem at heading - NSC content in the stem at maturity) × 100)/(NSC content in the stem at heading)

Nitrogen utilization efficiency for grain yield (NUEg) = (Rice grain yield)/(Total nitrogen accumulation)

Phosphorus utilization efficiency for grain yield (PUEg) = (Rice grain yield)/(Total phosphorus accumulation)

Potassium utilization efficiency for grain yield (KUEg) = (Rice grain yield)/(Total potassium accumulation)

Multifactorial analyses of variance were conducted to determine the effects of the year, treatment, and varietal type (as independent variables), as well as their interaction effects on the determined agronomic and physiological traits of rice (as dependent variables), at a significance level of 5%. Data were averaged over cultivation treatments or varietal types, when significant interaction effects were not observed among the year, cultivation treatment, and varietal types. Pairwise comparisons (using the Duncan test at a significance level of 5%) were also performed to compare the determined agronomic and physiological traits of rice. Pairwise comparisons showed no significant differences in any determined parameters among the three varieties in the same varietal type, so the data in the same varietal type were presented as the means of three varieties. All data analyses were conducted with SPSS 17.0 Software (SPSS Inc., Chicago, IL, USA).

## 3. Results

*3.1. Grain Yield and Its Components*

The differences in grain yield were significant ($p < 0.01$) regarding the cultivation treatment and varietal type. The 2010s' rice had 7.0% and 10.1% higher ($p < 0.05$) grain yield than the 1990s' rice under LCP and RNID across two years, respectively. It was also 3.6% and 5.5% higher ($p < 0.05$) than the 2000s' rice under LCP and RNID, respectively. Compared with LCP, the 2010s' rice had a similar grain yield, whereas the 1990s and 2000s' rice had a lower ($p < 0.05$) yield under RNID at two years. Generally, panicles per m$^2$, filled-grain percentage, and grain weight were decreased, whereas spikelets per panicle were increased with the rice varietal improvement. Compared with LCP, RNID raised panicles per m$^2$, whereas it reduced spikelets per panicle of three varietal types. For each varietal type, RNID increased the filled-grain percentage of the 2010s' rice ($p < 0.05$),

compared with LCP. The grain weight of the 1990s and the 2000s' rice were lower, although it was higher ($p < 0.05$) for the 2010s' rice under RNID than under LCP (Tables 3 and S1).

**Table 3.** Grain yield and its components of three varietal types under LCP and RNID.

| Year | Treatment | | Grain Yield (t ha$^{-1}$) | Panicles per m$^2$ | Spikelets per Panicle | Spikelets per m$^2$ | Filled-Grain Percentage (%) | Grain Weight (mg) |
|---|---|---|---|---|---|---|---|---|
| 2017 | Cultivation treatment | LCP | 11.1 a | 303 b | 155 a | 46.6 a | 89.4 a | 27.0 a |
| | | RNID | 10.9 a | 313 a | 146 b | 45.4 a | 89.9 a | 27.0 a |
| | Varietal type | 1990s | 10.6 c | 327 a | 131 c | 42.6 c | 90.0 a | 27.4 a |
| | | 2000s | 11.0 b | 311 b | 149 b | 46.3 b | 90.0 a | 27.2 a |
| | | 2010s | 11.5 a | 287 c | 172 a | 49.2 a | 88.9 a | 26.5 b |
| 2018 | Cultivation treatment | LCP | 11.1 a | 310 b | 153 a | 47.0 a | 88.6 a | 26.8 a |
| | | RNID | 10.9 b | 322 a | 143 b | 45.8 a | 89.9 a | 26.9 a |
| | Varietal type | 1990s | 10.6 c | 336 a | 135 c | 45.3 c | 90.3 a | 27.2 a |
| | | 2000s | 11.0 b | 319 a | 145 b | 46.2 b | 89.9 a | 26.8 ab |
| | | 2010s | 11.5 a | 293 b | 164 a | 47.8 a | 87.7 b | 26.6 b |

LCP, local cultivation practice; RNID, reduced nitrogen rate with increased planting density. Within each treatment, means followed by a different lower case letter are significant at the 5% level of significance.

### 3.2. Shoot Biomass Weight and Accumulation, Harvest Index, and NSC Content

There were differences ($p < 0.01$) in shoot biomass weight at maturity in cultivation treatment and varietal type; however, no differences ($p \geq 0.05$) in shoot biomass weight at jointing and heading were observed among year, cultivation treatment, varietal type, and their interactions. Compared with the 1990s and 2000s' rice, the 2010s' rice had more ($p < 0.05$) shoot biomass weight at heading and maturity, under both cultivation treatments; similar trends were detected for shoot biomass accumulation from jointing to heading and from heading to maturity. The 2010s' rice showed a higher harvest index relative to the 1990s' rice and the 2000s' rice under LCP and RNID. Compared with LCP, RNID reduced the shoot biomass weight at maturity of each varietal type, particularly for the 1990s and 2000s' rice ($p < 0.05$). The harvest index of the 2010s' rice was higher ($p < 0.05$) under RNID, compared with LCP (Tables 4 and S2).

**Table 4.** Shoot biomass weight and accumulation and harvest index of three varietal types under LCP and RNID.

| Year | Treatment | | Shoot Biomass Weight (t ha$^{-1}$) | | | Shoot Biomass Accumulation (t ha$^{-1}$) | | Harvest Index |
|---|---|---|---|---|---|---|---|---|
| | | | Jointing | Heading | Maturity | From Jointing to Heading | From Heading to Maturity | |
| 2017 | Cultivation treatment | LCP | 5.5 a | 11.9 a | 20.1 a | 6.5 a | 8.1 a | 0.476 b |
| | | RNID | 5.4 a | 11.7 a | 19.6 b | 6.3 a | 7.9 a | 0.484 a |
| | Varietal type | 1990s | 5.6 a | 11.7 a | 19.4 b | 6.1 b | 7.7 b | 0.471 b |
| | | 2000s | 5.4 a | 11.8 a | 19.8 b | 6.5 a | 8.0 b | 0.482 a |
| | | 2010s | 5.4 a | 12.0 a | 20.4 a | 6.6 a | 8.4 a | 0.488 a |
| 2018 | Cultivation treatment | LCP | 5.5 a | 11.7 a | 19.9 a | 6.2 a | 8.2 a | 0.481 b |
| | | RNID | 5.3 a | 11.4 a | 19.3 b | 6.1 a | 7.9 b | 0.489 a |
| | Varietal type | 1990s | 5.5 a | 11.4 a | 19.2 b | 5.9 a | 7.8 b | 0.477 b |
| | | 2000s | 5.5 a | 11.6 a | 19.5 b | 6.1 a | 7.9 b | 0.488 a |
| | | 2010s | 5.3 a | 11.8 a | 20.1 a | 6.5 a | 8.5 a | 0.491 a |

LCP, local cultivation practice; RNID, reduced nitrogen rate with increased planting density. Within each treatment, means followed by a different lower case letter are significant at the 5% level of significance.

Differences ($p < 0.01$) in NSC content in the stem were observed in the cultivation treatment and varietal types. The NSC content in the stem was increased with rice varietal improvement. Compared with LCP, RNID reduced ($p < 0.05$) the NSC content in the stem of each varietal type. The 2010's rice exhibited a lower NSC remobilization reserve under LCP, although it was higher ($p < 0.05$) under RNID than the 1990s and 2000s' rice over two cultivation years. Compared with LCP, RNID improved the NSC remobilization reserve of three varietal types, especially for the 2010's rice ($p < 0.05$) (Tables 5 and S3).

**Table 5.** NSC content in the stem and its remobilization reserve of three varietal types under LCP and RNID.

| Year | Treatment | | NSC Content in the Stem (g m$^{-2}$) | | NSC Remobilization Reserve (%) |
|---|---|---|---|---|---|
| | | | Heading | Maturity | |
| 2017 | Cultivation treatment | LCP | 309 a | 160 a | 48.2 b |
| | | RNID | 258 b | 115 b | 55.4 a |
| | Varietal type | 1990s | 250 b | 119 b | 52.4 a |
| | | 2000s | 281 b | 138 ab | 50.9 a |
| | | 2010s | 321 a | 157 a | 51.1 a |
| 2018 | Cultivation treatment | LCP | 311 a | 158 a | 49.2 b |
| | | RNID | 268 b | 121 b | 54.9 a |
| | Varietal type | 1990s | 258 b | 122 b | 52.7 a |
| | | 2000s | 291 b | 142 ab | 51.2 a |
| | | 2010s | 320 a | 155 a | 51.6 a |

NSC, non-structural carbohydrate; LCP, local cultivation practice; RNID, reduced nitrogen rate with increased planting density. Within each treatment, means followed by a different lower case letter are significant at the 5% level of significance.

### 3.3. LAI, SPAD Values, and Leaf Photosynthetic Rate

There were differences ($p < 0.01$) in the LAI at the main growth stages in terms of both cultivation treatment and varietal type. The varietal improvement increased the LAI at the main growth stages. For example, the 2010s' rice had a 22.1% and 12.2% higher LAI at heading, than the 1990s and 2000s' rice, respectively, under LCP. RNID caused larger reductions in LAI at the main growth stages for the 1990s' rice and the 2000s' rice, whereas there were relatively smaller reductions for the 2010s' rice, compared with LCP (Tables 6 and S4).

**Table 6.** LAI at the main growth stages of three varietal types under LCP and RNID.

| Year | Treatment | | LAI (m$^2$ m$^{-2}$) | | |
|---|---|---|---|---|---|
| | | | Jointing | Heading | Maturity |
| 2017 | Cultivation treatment | LCP | 4.1 a | 7.4 a | 2.5 a |
| | | RNID | 3.7 b | 7.2 a | 2.2 a |
| | Varietal type | 1990s | 3.3 b | 6.5 c | 1.9 c |
| | | 2000s | 4.1 a | 7.2 b | 2.4 b |
| | | 2010s | 4.4 a | 8.2 a | 2.8 a |
| 2018 | Cultivation treatment | LCP | 4.0 a | 7.5 a | 2.5 a |
| | | RNID | 3.6 a | 7.2 b | 2.2 a |
| | Varietal type | 1990s | 3.1 b | 6.7 c | 1.8 c |
| | | 2000s | 4.0 a | 7.3 b | 2.4 b |
| | | 2010s | 4.3 a | 8.2 a | 2.9 a |

LAI, leaf area index; LCP, local cultivation practice; RNID, reduced nitrogen rate with increased planting density. Within each treatment, means followed by a different lower case letter are significant at the 5% level of significance.

No differences ($p \geq 0.05$) in the SPAD values of the top three leaves among the three varietal types were detected at 12 DAH. The 2010s' rice had higher ($p < 0.05$) SPAD values of the top three leaves at 24 and 36 DAH, than the 1990s and 2000s' rice under two cultivation treatments. RNID caused lower reductions in the SPAD values of the top three leaves after the heading of the 2010s' rice, compared with the 1990s and 2000s' rice (Figures 1 and S1).

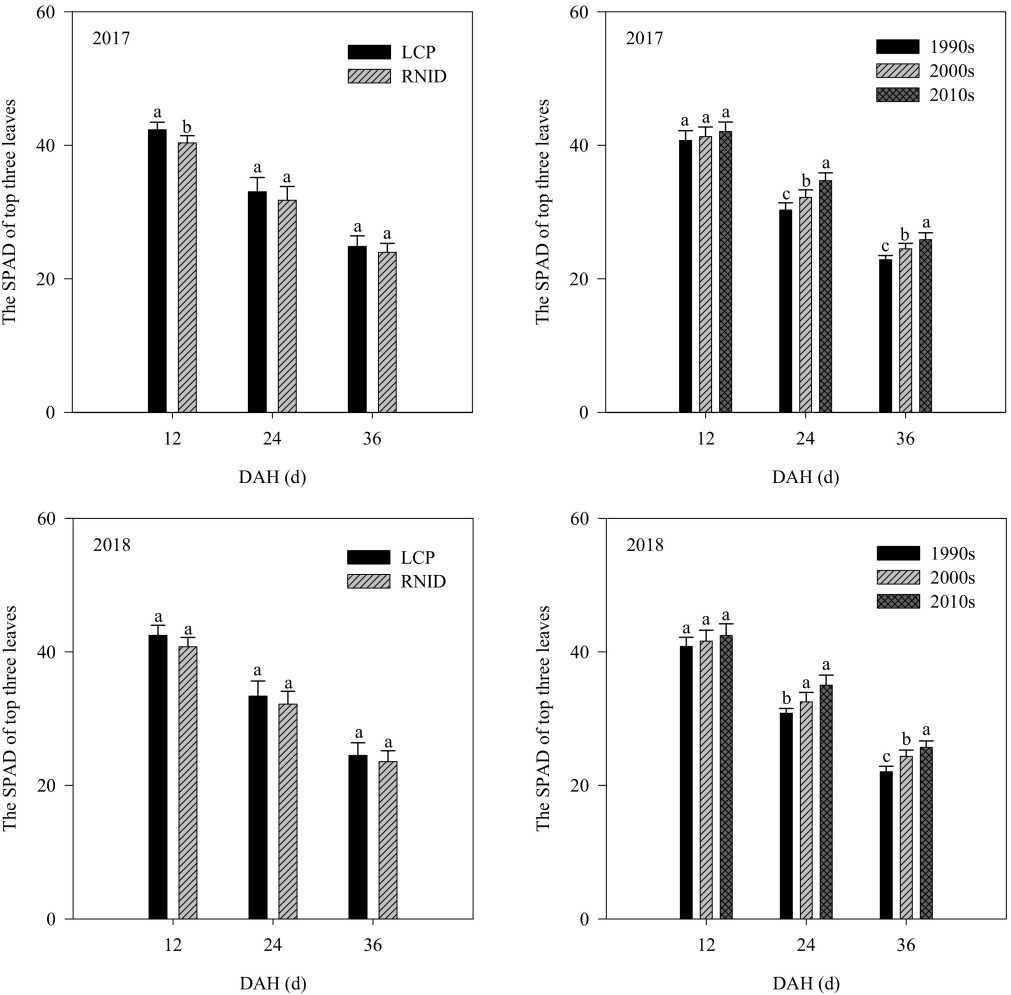

**Figure 1.** The SPAD of top three leaves at 12, 24, and 36 DAH between three varietal types and cultivation treatments. LCP, local cultivation practice; RNID, reduced nitrogen rate with increased planting density. DAH, days after heading. Within each treatment, bars marked with the different lower case letter indicate significance at the 5% level.

No differences ($p \geq 0.05$) in the photosynthetic rates of the flag leaves at 15 DAH were observed among the three varietal types under both cultivation treatments. The 2010s' rice showed a consistently higher ($p < 0.05$) flag leaf photosynthetic rate at 30 and 45 DAH than the 1990s and 2000s' rice under LCP and RNID. Compared with LCP, the flag leaf photosynthetic rate at 15, 30, and 45 DAH of the three varietal types were all lower under RNID (Figures 2 and S2).

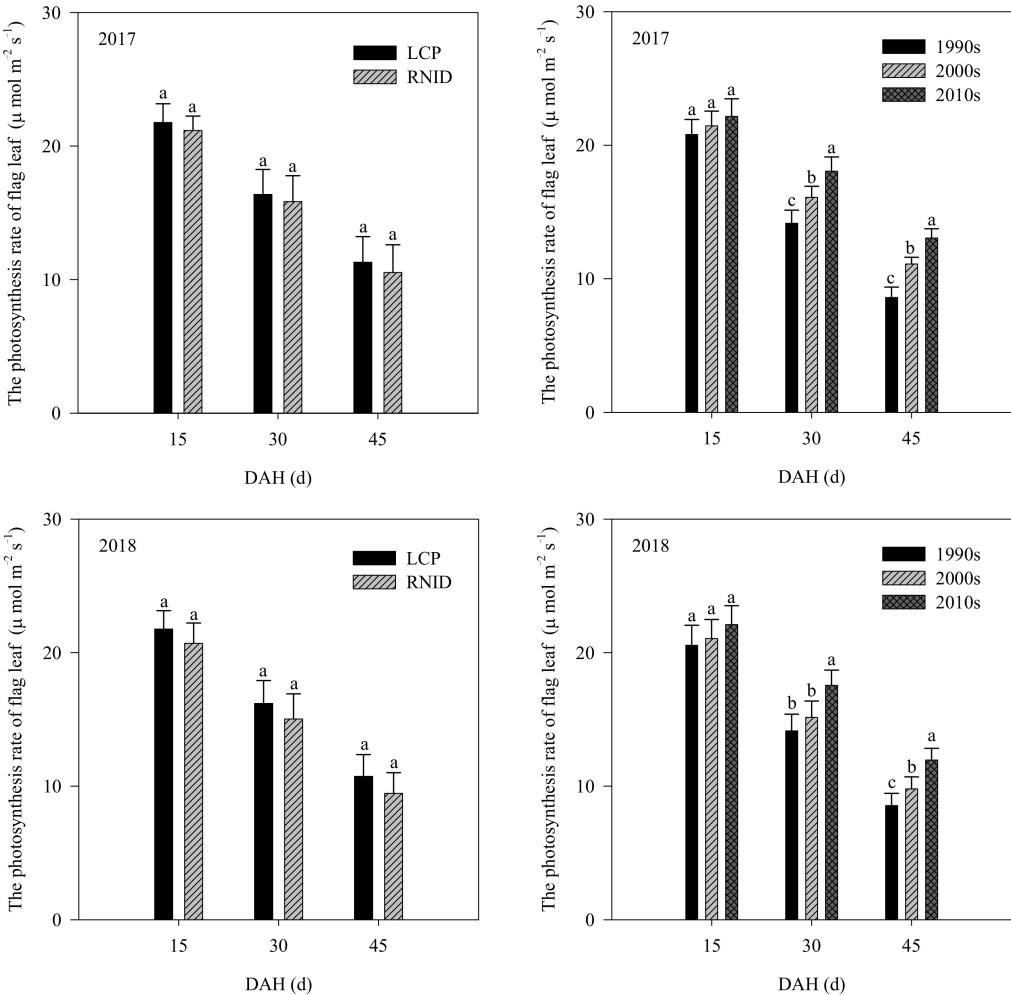

**Figure 2.** The photosynthetic rate of flag leaf at 15, 30, and 45 DAH between three varietal types and cultivation treatments. LCP, local cultivation practice; RNID, reduced nitrogen rate with increased planting density. DAH, days after heading. Within each treatment, bars marked with the different lower case letter indicate significance at the 5% level.

### 3.4. N, P, and K Accumulation and Their Utilization Efficiencies

The total N and P accumulations were varied ($p < 0.05$ or $p < 0.01$) in terms of cultivation treatment and varietal type, whereas the total K accumulation was only varied ($p < 0.01$) in varietal type. Compared with the 1990s and 2000s' rice, the 2010s' rice had a consistently higher total of N, P, and K accumulation, under both cultivation treatments. RNID reduced the total N accumulation of all three varietal types, especially for the 1990s and 2000s' rice, compared with LCP. Similarly, the total P and K accumulation of each varietal type was lower under RNID than under LCP. NUEg was varied ($p < 0.01$) in cultivation treatment and varietal type, whereas PUEg and KUEg were varied ($p < 0.05$ or $p < 0.01$) in varietal type. NUEg, PUEg, and KUEg were increased with varietal improvements. Compared with LCP, RNID increased the NUEg of the three varietal types, especially for the 2010s' rice ($p < 0.05$). Generally, the PUEg and KUEg of the 2010s' rice was higher under RNID than under LCP, whereas such results were not detected for the 1990s and 2000s' rice (Tables 7 and S5).

**Table 7.** Total N, P, and K accumulation and their use efficiency of three varietal types under LCP and RNID.

| Year | Treatment | | Total Nutrient Accumulation (kg ha$^{-1}$) | | | NUEg (kg Grain kg$^{-1}$) | PUEg (kg Grain kg$^{-1}$) | KUEg (kg Grain kg$^{-1}$) |
|------|-----------|---|---|---|---|---|---|---|
| | | | N | P | K | | | |
| 2017 | Cultivation treatment | LCP | 237 a | 60.8 a | 282 a | 46.8 b | 182 a | 39.4 a |
| | | RNID | 230 a | 59.3 a | 278 a | 47.7 a | 186 a | 39.5 a |
| | Varietal type | 1990s | 227 b | 58.6 b | 272 b | 46.8 a | 181 a | 38.9 b |
| | | 2000s | 233 b | 60.1 ab | 281 a | 47.4 a | 184 a | 39.3 b |
| | | 2010s | 242 a | 61.5 a | 287 a | 47.6 a | 188 a | 40.2 a |
| 2018 | Cultivation treatment | LCP | 236 a | 59.8 a | 285 a | 47.1 a | 186 a | 39.0 a |
| | | RNID | 227 b | 58.1 a | 280 a | 47.8 a | 187 a | 38.7 a |
| | Varietal type | 1990s | 227 b | 58.1 a | 278 b | 46.8 b | 183 a | 38.1 b |
| | | 2000s | 231 ab | 58.8 a | 282 ab | 47.4 ab | 187 a | 38.8 b |
| | | 2010s | 238 a | 60.1 a | 289 a | 48.2 a | 191 a | 39.8 a |

LCP, local cultivation practice; RNID, reduced nitrogen rate with increased planting density. N, nitrogen; P, phosphorus; K, potassium. NUEg, nitrogen utilization efficiency for grain yield; PUEg, phosphorus utilization efficiency for grain yield; KUEg, potassium utilization efficiency for grain yield. Within each treatment, means followed by a different lower case letter are significant at the 5% level of significance.

### 3.5. Correlation Analysis

Most determined parameters, such as the LAI at maturity, and the total N, P, and K accumulation, positively ($p < 0.01$ or $p < 0.05$) correlated with grain yield under both cultivation treatments. NSC negatively ($p < 0.01$) correlated with grain yield under LCP, although it correlated positively ($p < 0.01$) with grain yield under RNID (Table 8). There existed positive correlations between NUEg and the harvest index of rice under LCP and RNID. This was similar for PUEg and KUEg (Figure 3).

**Table 8.** Correlations between the determined parameters and grain yield of rice under LCP and RNID.

| The Determined Parameters | Grain Yield | |
|---|---|---|
| | LCP | RNID |
| Spikelets per panicle | 0.86 ** | 0.78 ** |
| Shoot biomass weight at maturity | 0.90 ** | 0.93 ** |
| Harvest index | 0.73 ** | 0.81 ** |
| Shoot biomass accumulation from heading to maturity | 0.64 ** | 0.68 ** |
| NSC remobilization reserve | −0.80 ** | 0.72 ** |
| LAI at maturity | 0.69 ** | 0.84 ** |
| The SPAD of top three leaves at 24 DAH | 0.76 ** | 0.74 ** |
| The SPAD of top three leaves at 36 DAH | 0.68 ** | 0.85 ** |
| The photosynthetic rate of flag leaf at 30 DAH | 0.77 ** | 0.91 ** |
| The photosynthetic rate of flag leaf at 45 DAH | 0.81 ** | 0.88 ** |
| Total N accumulation | 0.88 ** | 0.94 ** |
| Total P accumulation | 0.52 * | 0.57 * |
| Total K accumulation | 0.89 ** | 0.76 ** |
| NUEg | 0.59 * | 0.71 ** |
| PUEg | 0.52 * | 0.78 ** |
| KUEg | 0.55 * | 0.84 ** |

LCP, local cultivation practice; RNID, reduced nitrogen rate with increased planting density. NSC, non-structural carbohydrate; LAI, leaf area index; DAH, days after heading; N, nitrogen; P, phosphorus; K, potassium; NUEg, nitrogen utilization efficiency for grain yield; PUEg, phosphorus utilization efficiency for grain yield; KUEg, potassium utilization efficiency for grain yield. * significant at <0.05 level, and ** significant at <0.01 level.

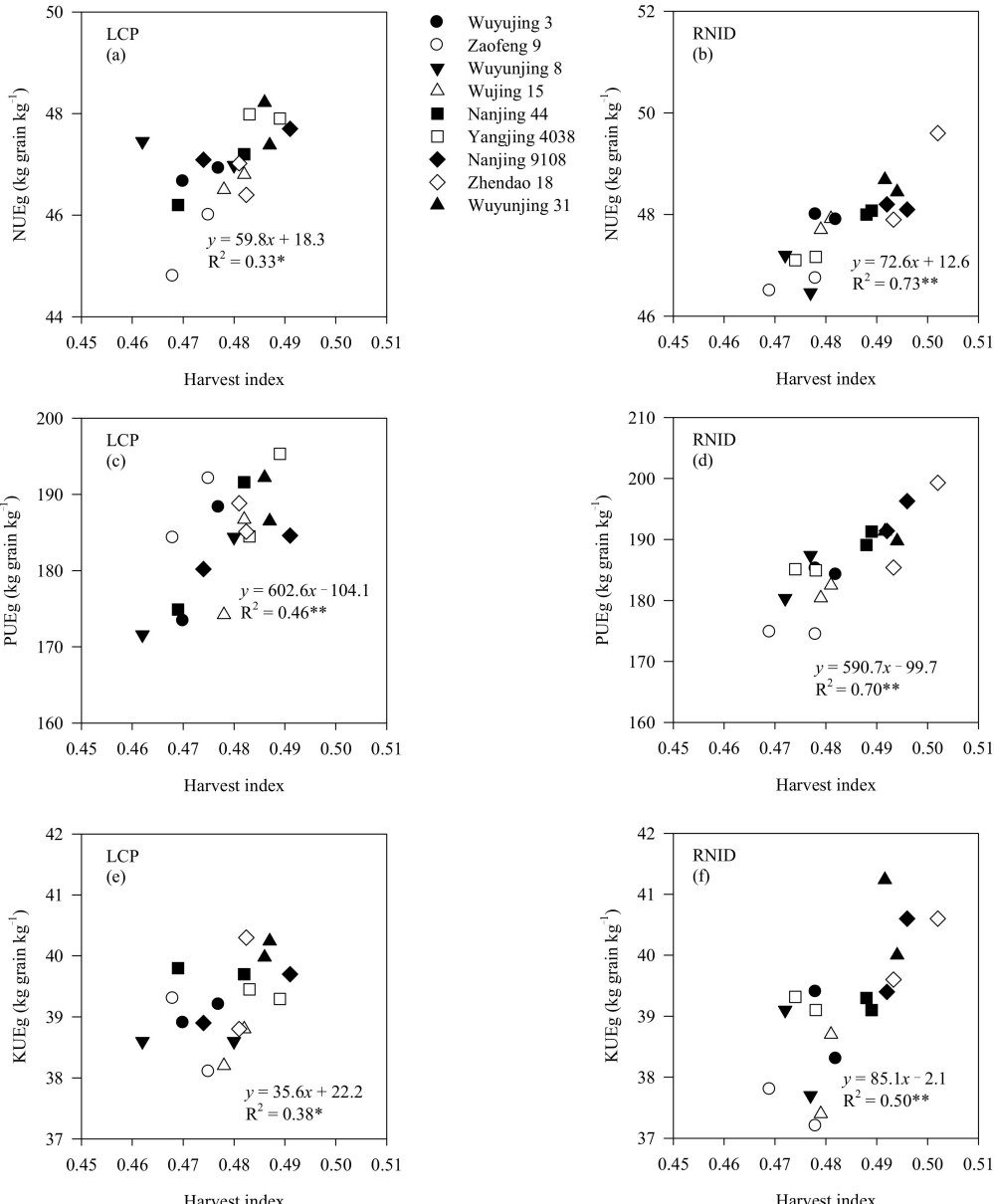

**Figure 3.** Correlations between harvest index and NUEg (**a**,**b**), PUEg (**c**,**d**), and KUEg (**e**,**f**) of rice under LCP and RNID. Data are collected from the rice varieties in 2017 and 2018. LCP, local cultivation practice; RNID, reduced nitrogen rate with increased planting density. NUEg, nitrogen utilization efficiency for grain yield; PUEg, phosphorus utilization efficiency for grain yield; KUEg, potassium utilization efficiency for grain yield. * significant at <0.05 level, and ** significant at <0.01 level.

## 4. Discussion

### 4.1. The Main Traits for the Progressive Yield Increase in Conventional japonica Rice since the 1990s

Similar to previous studies [5,6,8], the varietal improvement greatly increased the rice grain yield; the 2010s' rice had 3.6–7.0% and 5.5–10.1% higher ($p < 0.05$) grain yield, in relation to the 1990s and 2000s' rice under LCP and RNID, respectively (Tables 3 and S1). Our results indicated that a progressive yield gain was achieved during the genetic improvement of conventional *japonica* rice after the 1990s. The contribution of grain yield components to genetic yield gain was well documented, although the results were not entirely consistent. The yield increased during the genetic improvement as a result of a larger sink size through panicle per m$^2$ [6], spikelets per panicle [7,10], and a coordinated increase in panicles per m$^2$, spikelets per panicle, and filled-grain percentage [11]. Peng

et al. [8] concluded that the positive yield increase was not attributable to any single yield component. In this study, the 2010s' rice had more ($p < 0.05$) spikelets per panicle, despite having lower panicles per m$^2$, filled-grain percentage, and grain weight, compared with the 1990s and 2000s' rice (Table 3). Such results suggest that genetic progress in the grain yield was mainly driven by spikelets per panicle (Table 8), implying again that the increased spikelets per panicle is an efficient strategy for further yield increment in rice [32,35]. Moreover, a downward trend of sink-filling efficiency in the 2010s' rice, particularly filled-grain percentage, was detected during the genetic improvement (Table 3). Previous studies have also reported a relatively poor sink-filling problem in modern rice, with more spikelets per panicle [36,37]. This information implies that more attention should be paid to how to achieve synchronous improvements of sink size and sink-filling efficiency in modern large-panicle rice.

The existing literature has reported that rice grain yield increases due to shoot biomass, harvest index, or both [6–8] during genetic improvement. In this study, the 2000s' rice showed higher ($p < 0.05$) harvest index, despite a less significant increase in shoot biomass at maturity than the 1990's rice. The 2010s' rice had higher ($p < 0.05$) shoot biomass at maturity, although it had a less significant increase in harvest index than the 2000´s rice (Table 4). Our results indicated that the genetic yield gain from the 1990s to 2000s was mainly attributed to harvest index, although shoot biomass created the best yield for the 2000s to 2010s. With the varietal improvement, LAI at heading and maturity, SPAD values of the top three leaves at 24 and 36 DAH, and the flag leaf photosynthetic rate at 30 and 45 DAH were all increased (Table 5, Figures 1 and 2). Such results indicated that the stay-green ability of the shoot was greatly improved in the 2010s' rice, which also contributed to greater ($p < 0.05$) shoot biomass accumulation and grain yield than the 1990s and 2000s' rice (Tables 4 and 8).

The effects of the genetic improvement on N accumulation and NUE were studied in wheat, cotton, barley, and maize [16,18,20,21]. To date, such information is still limited for P and K elements in rice. In this study, the 2010s' rice exhibited a higher ($p < 0.05$) total N accumulation, relative to the 1990s and 2000s' rice, as well as for total P and K accumulation (Table 7). Moreover, NUEg, PUEg, and KUEg were all increased with varietal improvement (Table 7), indicating a great progress of nutrient use efficiency during the genetic process of conventional *japonica* rice in east China. Our results suggested that an increased grain yield also benefitted nutrient accumulation and their use efficiency (Table 8), similar to the results of Rochester and Constable [21] and Mueller et al. [18]. We also observed positive correlations between NUEg, PUEg, KUEg and the harvest index of rice (Figure 3), which implies that an improved nutrient use efficiency could be achieved by increasing the harvest index of crops [38].

### 4.2. The Feasibility of RNID Facilitating Grain Yield and NUE of Modern Conventional Japonica Rice

The 2010s' rice had a similar grain yield, whereas the 1990s and 2000s' rice had lower ($p < 0.05$) grain yields under RNID, relative to LCP (Table S1). In addition, the NUEg of the 2010s' rice was also increased ($p < 0.05$) under RNID (Table S5). These results indicate that RNID facilitated grain yield and NUE in modern conventional *japonica* rice rather than the old ones, which was consistent with the studies conducted with hybrid *indica* rice in south and central China [23,25,27]. For three varietal types, the increased panicles per m$^2$ could not fully compensate for the reduction of spikelets per panicle, which resulted in lower sink size under RNID. It is noteworthy that the filled-grain percentage and grain weight of the 2010s' rice increased ($p < 0.05$) under RNID, although this was not consistent for the 1990s and 2000s' rice, compared with LCP (Table S1). Our results indicated that higher panicles per m$^2$, and an improved sink-filling efficiency maintained the grain yield of modern conventional *japonica* rice under RNID.

Compared with LCP, RNID reduced LAI, as well as the SPAD values of the top three leaves and flag leaf photosynthetic rate after heading, of each variety type (Table S4, Figures S1 and S2). However, such reductions were smaller in the 2010s' rice in relation

to the 1990s and 2000s' rice, which might be linked to improved shoot stay-green traits of the 2010s' rice, as discussed above. The reduced leaf-related properties also resulted in a lower shoot biomass weight of the three varietal types under RNID. Nevertheless, the harvest index was increased under RNID, particularly for the 2010s' rice ($p < 0.05$). This result suggests that the increased harvest index of the 2010s' rice under RNID facilitated a grain yield and NUE relative to LCP (Tables S1 and S4, Figure 3), consistent with prior studies [22,26,29].

Sink-filling efficiency and the harvest index of the 2010s' rice were increased under RNID (Tables S1 and S2); it is reported that an improved remobilization of NSC from source to sink benefitted grain filling and the harvest index [37,39]. Compared with LCP, RNID reduced NSC content and increased the NSC remobilization reserve, especially for the 2010s' rice (Tables 5 and S3). Therefore, for the 2010s' rice, the better remobilization of NSC after heading was helpful to increase the harvest index, sink-filling efficiency, grain yield, and NUE.

## 5. Conclusions

Genetic improvements have greatly increased the grain yield and nitrogen use efficiency of conventional *japonica* rice after the 1990s in East China. Relative to the 1990s' rice and the 2000s' rice, the 2010s' rice had a 3.6–7.0% higher grain yield under LCP, and 5.5–10.1% higher under RNID. The genetic yield gain from the 1990s to 2000s was mainly attributed to the harvest index, whereas shoot biomass contributed more to the increased yield for the 2000s to 2010s. Genetic selection for grain yield also increased the total N, P, and K accumulation, and their use efficiencies. A higher nutrient use efficiency could be achieved by increasing the harvest index in rice. RNID maintained the grain yield and NUE of modern, conventional *japonica* rice, and attributed its improved sink-filling efficiency, post-heading NSC remobilization, and harvest index.

**Supplementary Materials:** The following are available online at https://www.mdpi.com/article/10.3390/agriculture11121188/s1, Table S1: Grain yield and its components of three varietal types under LCP and RNID, Table S2: Shoot biomass weight and accumulation, and harvest index of three varietal types under LCP and RNID, Table S3: NSC content in the stem and its remobilization reserve of three varietal types under LCP and RNID, Table S4: LAI at the main growth stages of three varietal types under LCP and RNID, Figure S1: The SPAD of top three leaves at 12, 24, and 36 DAH of three varietal types under LCP and RNID, Figure S2: The photosynthetic rate of flag leaf at 15, 30, and 45 DAH of three varietal types under LCP and RNID, Table S5: Total N, P, and K accumulation and their use efficiency of three varietal types under LCP and RNID.

**Author Contributions:** Conceptualization, T.M.; experimental design, Q.D. and H.W.; investigation, X.C., J.G. and X.Z.; data curation, G.Z.; writing—original draft preparation, T.M.; funding acquisition, Q.D. and H.W. All authors have read and agreed to the published version of the manuscript.

**Funding:** This work was financed by the Key Research and Development Program of Jiangsu Province (BE2019343), National Natural Science Foundation of China (31901448, 32001466), Postdoctoral Research Foundation of China (2020M671628, 2020M671629), and Joints Funds of the National Natural Science Foundation of China (U20A2022).

**Institutional Review Board Statement:** Not applicable.

**Informed Consent Statement:** Not applicable.

**Data Availability Statement:** Not applicable.

**Conflicts of Interest:** The authors declare no conflict of interest.

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
