# Peer review of "Reduced Nitrogen Rate with Increased Planting Density Facilitated Grain Yield and Nitrogen Use Efficiency in Modern Conventional Japonica Rice"

_agriculture, doi:10.3390/agriculture11121188_

Round 1

Reviewer 1 Report

  • Concise the abstract to 200 words
  • Line 22: Change ‘released in’ to ‘released during’
  • Line 30-32: Reframe as ‘For three varietal types, RNID increased panicles per m2, filled-grain percentage and grain weight (p < 0.05) while decreased spikelets per panicle of the 2010’s rice.’
  • Line 33: Correct non-structural carbohydrate (NSC) content
  • Line 34: Delete impressive
  • Line 37: Reframe sentence to ‘while from the 2000s to 2010s, yield increase contributed through shoot biomass’
  • Line 39-40: Arrange keywords in alphabetical order
  • Line 86 Reframe sentence as ‘To investigate the possibility of RNID facilitated changes in grain yield and nutrient use efficiency of conventional japonica rice in Jiangsu.’
  • Line 93-94: The mean climatic conditions (such as temperature, rainfall, humidity, and sunshine hours) should be mentioned.
  • Line 107: Reframe sentence as ‘The selection of these rice varieties was based on cultivation by local institutes and popularity among farmers.’
  • Line 108: Reframe ‘The planting area ………east China.’ sentence to ‘ For example, Wuyujing 3 variety was most popular variety in the 1990s in east China and its cultivation area was up to 6.1 million hectares by end of 2019.’
  • Line 109: Reframe line ‘In recent years……………….rice production’ to ‘In recent years, Nanjing 9108 and Wuyunjing 13 become very popular and widely planted rice varieties owing to their superior palatability.’
  • Line 111: Delete beside
  • Line 116: Delete full stop in the title of all tables mentioned in the manuscript.
  • Line 117: Write ‘Source’ instead of ‘Such information is available from the’
  • Line 118: Pre-sowing treatment of rice seeds should be mentioned
  • Line 119: Rephrase ‘12 June…..two years.’ to ‘12 June during the rice cultivation season of 2017 and 2018’
  • Line 122: Delete ‘both’
  • Line 123-124: Mention whether chemical or organic methods used for pest, disease, and weed management in the study
  • Line 136: Expand SPAD
  • Line 137-138: Why different days were chosen for SPAD and Leaf photosynthetic index.
  • Line 138-139: Delete ‘soil-plant analysis development meter’ and un-bracket SPAD-502 plus meter.
  • Line 148: Mention reference for the method.
  • Line 167-169: Reframe the line to increase understandability. Kindly explain how you calculated the percent increase.
  • Combine Table 2 with 3; Table 4 with 5 and 6; Table 6 with 7
  • Line 200: Rephrase ‘rice at two years’ as ‘rice during two cultivation years’
  • Table 4, 5, and 6: Un-bold the data in the table
  • Line 384: Follow the journal’s  rule for citing references in article  

Author Response

Point 1: Concise the abstract to 200 words

Response 1: As suggested, we simplified the Abstract as follows: “The past three decades have seen a pronounced development of conventional japonica rice after the 1990s, whereas little information is available on changes regarding grain yield and nutrient use efficiency during this process. Nine conventional japonica rice released during the 1990s, 2000s, and 2010s were grown under reduced nitrogen rate with increased planting density (RNID) and local cultivation practice (LCP) in 2017 and 2018. The 2010’s rice had 3.6%-5.5% and 7.0%-10.1% higher (p < 0.05) grain yield than the 2000s’ and the 1990s’ rice, respectively, under RNID and LCP. Harvest index contributed more to genetic yield gain from the 1990s to 2000s; while from the 2000s to 2010s, yield increase contributed through shoot biomass. Genetic improvement increased total nitrogen (N), phosphorus (P), and potassium (K) accumulation, and their use efficiencies. The 2010’s rice showed a similar grain yield, while the 1990s’ rice and the 2000s’ rice exhibited lower (p < 0.05) grain yield under RNID relative to LCP. RNID increased N, P, and K use efficiencies, particularly for N use efficiency for grain yield (NUEg) of the 2010’s rice, compared with LCP. For three varietal types, RNID increased panicles per m2, filled-grain percentage and grain weight (p < 0.05) while decreased spikelets per panicle of the 2010’s rice. Compared with LCP, RNID reduced non-structural carbohydrate content (NSC) and shoot biomass at heading and maturity, while increased remobilization of NSC and harvest index, especially for the 2010’s rice. Our results suggested the impressive progressive increase of grain yield and nutrient use efficiency of conventional japonica rice since the 1990s in east China. RNID could facilitate grain yield and NUEg of modern conventional japonica rice, rather than the old ones. ”

Point 2: Line 22: Change ‘released in’ to ‘released during’

Response 2: Good suggestion. We changed ‘released in’ to ‘released during’ in Lines 17 and 79 in the revised manuscript (clean version).

Point 3: Line 30-32: Reframe as ‘For three varietal types, RNID increased panicles per m2, filled-grain percentage and grain weight (< 0.05) while decreased spikelets per panicle of the 2010’s rice.’

Response 3: Good suggestion. We reframed as suggested in Lines 26-28 in the revised manuscript (clean version).

Point 4: Line 33: Correct non-structural carbohydrate (NSC) content

Response 4: Good suggestion. We corrected changed ‘non-structural carbohydrate (NSC) content’ in Line 29 in the revised manuscript (clean version).

Point 5: Line 34: Delete impressive

Response 5: Good suggestion. We deleted ‘impressive’ in the revised manuscript.

Point 6: Line 37: Reframe sentence to ‘while from the 2000s to 2010s, yield increase contributed through shoot biomass’

Response 6: Good suggestion. We reframed as suggested in Lines 21-22 in the revised manuscript (clean version).

Point 7: Line 39-40: Arrange keywords in alphabetical order

Response 7: Good suggestion. We re-arranged keywords in alphabetical order in the revised manuscript.

Point 8: Line 86 Reframe sentence as ‘To investigate the possibility of RNID facilitated changes in grain yield and nutrient use efficiency of conventional japonica rice in Jiangsu.’

Response 8: Good suggestion. We reframed as suggested in Lines 82-83 in the revised manuscript (clean version).

Point 9: Line 93-94: The mean climatic conditions (such as temperature, rainfall, humidity, and sunshine hours) should be mentioned.

Response 9: Good suggestion. We added the data of mean temperature, rainfall, and sunshine hours of rice growing-period in 2017 and 2018. Such information was seen in Lines 88-89 in the revised manuscript (clean version).

Point 10: Line 107: Reframe sentence as ‘The selection of these rice varieties was based on cultivation by local institutes and popularity among farmers.’

Response 10: Good suggestion. We reframed as suggested in Lines 107-108 in the revised manuscript (clean version).

Point 11: Line 108: Reframe ‘The planting area ………east China.’ sentence to ‘ For example, Wuyujing 3 variety was most popular variety in the 1990s in east China and its cultivation area was up to 6.1 million hectares by end of 2019.’

Response 11: As suggested, we changed as follows: “For example, Wuyujing 3 variety was most popular variety in the 1990s in east China and its cultivation area was up to 6.1 million hectares by end of 2019 (www.ricedata.cn).” in Lines 109-110 in the revised manuscript (clean version).

Point 12: Line 109: Reframe line ‘In recent years……………….rice production’ to ‘In recent years, Nanjing 9108 and Wuyunjing 13 become very popular and widely planted rice varieties owing to their superior palatability.’

Response 12: Good suggestion. We reframed as follows: “In recent years, Nanjing 9108 and Wuyunjing 13 become very popular and widely planted rice varieties owing to their superior palatability [13].” in Lines 110-112 in the revised manuscript (clean version).

Point 13: Line 111: Delete beside

Response 13: Good suggestion. We deleted ‘besides’ in the revised manuscript.

Point 14: Line 116: Delete full stop in the title of all tables mentioned in the manuscript.

Response 14: As suggested, we deleted full stop in the title of all tables mentioned in the revised manuscript.

Point 15: Line 117: Write ‘Source’ instead of ‘Such information is available from the’

Response 15: Good suggestion. We changed as follows: “Source: http://www.ricedata.cn.” in Line 117 in the revised manuscript (clean version).

Point 16: Line 118: Pre-sowing treatment of rice seeds should be mentioned

Response 16: As suggested. Such information was revised as: “Pre-germinated rice seeds were sown in seedbeds on 23 May” in Line 118 in the revised manuscript (clean version).

Point 17: Line 119: Rephrase ‘12 June…..two years.’ to ‘12 June during the rice cultivation season of 2017 and 2018’

Response 17: Good suggestion. We revised as suggested in Line 119 in the revised manuscript (clean version).

Point 18: Line 122: Delete ‘both’

Response 18: Good suggestion. We deleted ‘both’ in the revised manuscript.

Point 19: Line 123-124: Mention whether chemical or organic methods used for pest, disease, and weed management in the study

Response 19: Good suggestion. The chemical methods were applied for controlling pest, disease, and weed in the study. Such information was revised as follows: “Chemical control of pest, disease, and weed were performed following local recommendations.” in Lines 124-125 in the revised manuscript (clean version).

Point 20: Line 136: Expand SPAD

Response 20: We expanded SPAD as suggested. Such information was follows: “soil–plant analysis development (SPAD) values of nine rice varieties were determined at 12, 24, and 36 days after heading (DAH)” in Lines 137-138 in the revised manuscript (clean version).

Point 21: Line 137-138: Why different days were chosen for SPAD and Leaf photosynthetic index.

Response 21: Good suggestion. The different days chosen for measuring SPAD and Leaf photosynthetic index was mainly considering such measurements were particularly laborious. Therefore, we selected the different days for SPAD and Leaf photosynthetic index.

Point 22: Line 138-139: Delete ‘soil-plant analysis development meter’ and un-bracket SPAD-502 plus meter.

Response 22: Revised as suggested.

Point 23: Line 148: Mention reference for the method.

Response 23: Good suggestion. We added the reference for measuring filled-grain percentage in Lines 147-149 in the revised manuscript (clean version).

Point 24: Line 167-169: Reframe the line to increase understandability. Kindly explain how you calculated the percent increase.

Response 24: Good suggestion. We revised as follows: “The 2010s’ rice had 7.0% and 10.1% higher (p < 0.05) grain yield than the 1990s’ rice under LCP and RNID across two years, respectively; and 3.6% and 5.5% higher (p < 0.05) than the 2000s’ rice under LCP and RNID, respectively.” in Lines 173-175 in the revised manuscript (clean version).

The percent increase was calculated as follows, for example, the 2010s’ rice had 7.0% higher (p < 0.05) grain yield than the 1990s’ rice under LCP across two years. Here, 7.0%=( - )×100/( ).

Point 25: Combine Table 2 with 3; Table 4 with 5 and 6; Table 6 with 7

Response 25: Combined with the comments from Reviewer 2, we re-arranged the Tables in the revised manuscript. The original Table 2 and 3 as new Table S1, and Table 7 and 8 as new Table S5 in the revised manuscript.

Point 26: Line 200: Rephrase ‘rice at two years’ as ‘rice during two cultivation years’

Response 26: Good suggestion. We revised as suggested in Lines 203-204 in the revised manuscript (clean version).

Point 27: Table 4, 5, and 6: Un-bold the data in the table

Response 27: Revised as suggested.

Point 28: Line 384: Follow the journal’s rule for citing references in article  

Response 28: As suggested. We checked this citation following the journal’s rule and changed as follows: “Breseghello, F.; De Morais, O.P.; Pinheiro, P.V.; Silva, A.C.S.; De Castro, E.D.M.; Guimarães, É.P.; De Castro, A.P.; Pereira, J.A.; Lopes, A.D.M.; Utumi, M.M.; De Oliveira, J.P. Results of 25 years of upland rice breeding in Brazil. Crop Sci. 2011, 51, 914-923.” in Line in the revised manuscript (clean version).

Thank you for your professional comments and precious time to improve this manuscript.

Reviewer 2 Report

The manuscript presents interesting study on possibilities of the reduction of nitrogen fertilization, especially because of increase of nitrogen fertilizers prices.

Description of experimental design is not clear. How the replicates were randomized?

Was the design complete? It should be clearly stated how many plots in total were conducted and if all combinations of year x cultivation treatment x varietal type were observed. It is not clear.

It would be good if the differences between LCP and RNID were presented. I propose to show these differences in new table in section 2.1.

Line 158: What procedure of multiple comparisons was used? It LSD at 0.05 probability level? Please provide more details. Please notice that Fishers’s LSD is not recommended if more than two means are compared.

Because the main aim of the study was comparison between LCP and RNID I recommend add marginal means for these two treatments to present the main effect of the treatments (general difference between these two treatments) for all studied traits (results presented in all tables and in all figures).

Table 8: Please use superscripts (for “-1”) and subscripts (for “0.05” after LSD).

References are not properly listed. The distance between numbers and the text of the references is too large.

Author Response

Point 1: Description of experimental design is not clear. How the replicates were randomized?

Response 1: Good suggestion. We revised as follows: “The field experiment was a 2 × 9 (two cultivation treatments and nine rice varieties) factorial design with 18-treatments combinations.” in Lines 93-94 in the revised manuscript (clean version).

Point 2: Was the design complete? It should be clearly stated how many plots in total were conducted and if all combinations of year x cultivation treatment x varietal type were observed. It is not clear.

Response 2: As suggested, we revised such information as follows: “The field experiment was a 2 × 9 (two cultivation treatments and nine rice varieties) factorial design with 18-treatments combinations. Each of the treatment had three replications in a complete randomized block design, and gave a total of 54 plots.” in Lines 93-95 in the revised manuscript (clean version).

Point 3: It would be good if the differences between LCP and RNID were presented. I propose to show these differences in new table in section 2.1.

Response 3: Good suggestion. We added a new Table about information of nitrogen management and planting density of LCP and RNID in the revised manuscript (clean version).

Point 4: Line 158: What procedure of multiple comparisons was used? It LSD at 0.05 probability level? Please provide more details. Please notice that Fishers’s LSD is not recommended if more than two means are compared.

Response 4: Good suggestion. Good suggestion. We seek the help of Prof. Xu (Prof. Chenwu Xu, a specialist in data analysis) in our university about the statistical analysis in this study. Followed with the suggestions of Prof. Xu, we changed the description in statistical analysis, and such information was follows in Lines 159-169 (clean version): “Multivariate analyses of variance (MANOVA) were conducted to determine the effects of year, treatment, and varietal type (as independent variables) as well as their interaction effects on the determined agronomic and physiological traits of rice (as dependent variables) at a significance level of 5%. Data were averaged over cultivation treatments or varietal types when significant interaction effects were not observed among year, cultivation treatment, and varietal types. Pairwise comparisons (using Duncan test at a significance level of 5%) were also performed to compare the determined agronomic and physiological traits of rice. Pairwise comparisons showed no significant differences in any determined parameters among the three varieties in the same varietal type, so the data in the same varietal type were presented as the means of three varieties. All data analyses were conducted with SPSS 17.0 Software (SPSS Inc., Chicago, USA).”

Point 5: Because the main aim of the study was comparison between LCP and RNID. I recommend add marginal means for these two treatments to present the main effect of the treatments (general difference between these two treatments) for all studied traits (results presented in all tables and in all figures).

Response 5: As suggested, we re-arranged the Tables and Figures in the revised manuscript. The original Table 3-7 was listed as Table S1-S5, and Figures 1 and 2 as Figure S1 and S2.

Point 6: Table 8: Please use superscripts (for “-1”) and subscripts (for “0.05” after LSD).

Response 6: Revised as suggested.

Point 7: References are not properly listed. The distance between numbers and the text of the references is too large.

Response 7: Revised as suggested.

Thank you for your professional comments and precious time to improve this manuscript.

Round 2

Reviewer 2 Report

The manuscript was improved according all my comments. I have only one doubt. In Material and methods there is information that: "Multivariate analyses of variance (MANOVA) were conducted ..." but in the manuscript only results for univariate analyses are presented. In my opinion this description should be replaced by: "Multifactorial analyses of variance were conducted..." because effect of many factors was examined by for each variable separately not for sets of variables.

Author Response

Point 1: The manuscript was improved according all my comments. I have only one doubt. In Material and methods there is information that: "Multivariate analyses of variance (MANOVA) were conducted ..." but in the manuscript only results for univariate analyses are presented. In my opinion this description should be replaced by: "Multifactorial analyses of variance were conducted..." because effect of many factors was examined by for each variable separately not for sets of variables.

Response 1: Good suggestion. We revised as follows: “Multifactorial analyses of variance were conducted to determine the effects of year, treatment, and varietal type (as independent variables) as well as their interaction effects on the determined agronomic and physiological traits of rice (as dependent variables) at a significance level of 5%.” in Lines 159-162 in the revised manuscript (clean version).

Thank you for your professional comments and precious time to improve this manuscript.
